# Age-related differences in the functional topography of the locus coeruleus and their implications for cognitive and affective functions

**Dániel Veréb[1]\*, Mite Mijalkov[1], Anna Canal-Garcia[1], Yu-Wei Chang[2], Emiliano Gomez-Ruiz[2], Blanca Zufiria Gerboles[1], Miia Kivipelto[1,3], Per Svenningsson[3,4], Henrik Zetterberg[5,6,7,8,9,10], Giovanni Volpe[2], Matthew Betts[11,12,13], Heidi IL Jacobs[14,15], Joana B Pereira[1,16]\***

[1]Department of Neurobiology, Care Sciences and Society, Division of Clinical Geriatrics, Karolinska Institutet, Stockholm, Sweden; [2]Department of Physics, Goteborg University, Goteborg, Sweden; [3]University of Eastern Finland, Kuopio, Finland; [4]Department of Clinical Neuroscience, Karolinska Institutet, Stockholm, Sweden; [5]Department of Psychiatry and Neurochemistry, Institute of Neuroscience and Physiology, the Sahlgrenska Academy at the University of Gothenburg, Mölndal, Sweden; [6]Clinical Neurochemistry Laboratory, Sahlgrenska University Hospital, Mölndal, Sweden; [7]Department of Neurodegenerative Disease, UCL Institute of Neurology, London, United Kingdom; [8]UK Dementia Research Institute at UCL, London, United Kingdom; [9]Hong Kong Center for Neurodegenerative Diseases, Clear Water Bay, Hong Kong, China; [10]Wisconsin Alzheimer's Disease Research Center, University of Wisconsin School of Medicine and Public Health, University of Wisconsin-Madison, Madison, United States; [11]Institute of Cognitive Neurology and Dementia Research (IKND), Otto-von-Guericke University Magdeburg, Magdeburg, Germany; [12]German Center for Neurodegenerative Diseases (DZNE), Otto-von-Guericke University Magdeburg, Magdeburg, Germany; [13]Center for Behavioral Brain Sciences, University of Magdeburg, Magdeburg, Germany; [14]Maastricht University, Maastricht, Netherlands; [15]Massachusetts General Hospital, Boston, United States; [16]Memory Research Unit, Department of Clinical Sciences Malmö, Lund University, Lund, Sweden

**\*For correspondence:**
daniel.vereb@ki.se (DV);
joana.pereira@ki.se (JBP)

**Abstract** The locus coeruleus (LC) is an important noradrenergic nucleus that has recently attracted a lot of attention because of its emerging role in cognitive and psychiatric disorders. Although previous histological studies have shown that the LC has heterogeneous connections and cellular features, no studies have yet assessed its functional topography in vivo, how this heterogeneity changes over aging, and whether it is associated with cognition and mood. Here, we employ a gradient-based approach to characterize the functional heterogeneity in the organization of the LC over aging using 3T resting-state fMRI in a population-based cohort aged from 18 to 88 years of age (Cambridge Centre for Ageing and Neuroscience cohort, n=618). We show that the LC exhibits a rostro-caudal functional gradient along its longitudinal axis, which was replicated in an independent dataset (Human Connectome Project [HCP] 7T dataset, n=184). Although the main rostro-caudal direction of this gradient was consistent across age groups, its spatial features varied with increasing age, emotional memory, and emotion regulation. More specifically, a loss of rostral-like

connectivity, more clustered functional topography, and greater asymmetry between right and left LC gradients was associated with higher age and worse behavioral performance. Furthermore, participants with higher-than-normal Hospital Anxiety and Depression Scale (HADS) ratings exhibited alterations in the gradient as well, which manifested in greater asymmetry. These results provide an in vivo account of how the functional topography of the LC changes over aging, and imply that spatial features of this organization are relevant markers of LC-related behavioral measures and psychopathology.

## eLife assessment

This study provides **fundamental** imaging evidence from two independent functional imaging datasets, for a rostral-caudal gradient of locus coeruleus connectivity, which changes across the lifespan. The gradient approach is well-established and **convincing** results were obtained and validated using large 3T and 7T fMRI datasets. This work will be of interest to clinical and cognitive neuroscientists.

## Introduction

The central noradrenergic system comprises projections to widespread cortical and subcortical areas of the brain, with important neuromodulatory roles in sleep and arousal, executive function, mood, emotion regulation, and memory (*Samuels and Szabadi, 2008a*). Most of the noradrenergic transmission in the brain originates from the locus coeruleus (LC), a bilateral nucleus in the dorsolateral pons. The LC has been implicated in aging-related cognitive changes and neurodegenerative conditions, such as Alzheimer's disease (AD) or Parkinson's disease (PD) (*Betts et al., 2019b*). Interestingly, different conditions seem to affect different parts of the LC (*Madelung et al., 2022*; *Ye et al., 2022*), and several studies show that subregions of the LC might be involved in different cognitive functions and psychiatric symptoms (*Hämmerer et al., 2018*; *Ciampa et al., 2022*). Specifically, the rostral part is associated with memory and emotion regulation, being often affected in AD (*Hämmerer et al., 2018*; *Ciampa et al., 2022*; *Betts et al., 2019a*; *Theofilas et al., 2017*), whereas the integrity of the caudal part has been linked to more general cognitive processes in PD (*Ye et al., 2022*; *Doppler et al., 2021*).

However, a dichotomous functional division model of the LC might be oversimplistic, and technical difficulties concerning imaging of the LC (such as low spatial resolution of functional imaging techniques, inconvenient anatomical location, or noise from adjacent cerebrospinal fluid spaces; *Engels-Domínguez et al., 2023*) could hide a more complex underlying functional organization. Additionally, multiple overlapping axes of functional organization might be present that can confound current traditional cluster-based models that seek to divide the LC into subregions (*Haak and Beckmann, 2020*; *Poe et al., 2020*). Similar issues have been raised regarding other subcortical nuclei, where researchers started to unravel this complexity by modeling voxel-wise functional organization with a gradient-based approach. For example, several studies reported that the hippocampus is organized by a gradient of connectivity profiles and microstructural properties along its longitudinal axis (*Przeździk et al., 2019*). These spatial characteristics of hippocampal gradients explained individual differences in behavior better than a cluster-based model, and therefore provided a closer approximation to the true organization profile of this region. Other studies investigated gradients of functional connectivity in the striatum in healthy individuals and patients with PD. Oldehinkel et al. found that, apart from the primary functional connectivity gradient, a second axis of connectopic organization was present in the striatum that closely resembled the spatial distribution of dopamine transporters across different medication and symptomatology stages, providing a non-invasive marker for striatal dopamine transporter density in PD (*Oldehinkel et al., 2022*).

To our knowledge, so far, no studies have investigated the functional gradients of the LC. This is important because this region modulates various behavioral functions and widespread brain regions in a very precise manner (*Liu et al., 2020*). Furthermore, studies using fMRI to describe connectivity in the LC report heterogeneous results – possibly because the localization methods, the studied populations, and preprocessing pipelines often differ (*Liebe et al., 2020*; *Song et al., 2021*; *Liebe et al., 2022*; *Zhang et al., 2016*) – and a similarity gradient-based approach might be better suited to characterize its functional organization by mitigating the effect of heterogeneous individual differences

**Table 1.** Demographic features of the CamCAN and Human Connectome Project (HCP) 7T cohorts. Values in the table represent means followed by (standard deviations), except when stated otherwise. Measures from the emotional memory task represent d prime scores, which demonstrate the sensitivity of an individual to detect or correctly categorize stimuli in the given condition. Emotion regulation measures represent rescaled Likert-scale ratings, whereas the Hotel-task is scored as the average time in seconds a participant spends on each subtask. PSQI is a self-reported measure of sleep quality. HADS is a screening tool for anxiety and depression symptoms. Abbreviations: SD=standard deviation, M=male, F=female, PSQI=Pittsburgh Sleep Quality Index, HADS=Hospital Anxiety and Depression Scale.

|  | CamCAN (n=618) | HCP 7T (n=184) |
| --- | --- | --- |
| *Age range (years)* | 18–88 | 22–36 |
| *Biological sex (M/F)* | 305/313 | 72/112 |
| *Education years* | 14.56 (4.02) | – |
| *Emotional memory – recollection* | 1.58 (0.69) | – |
| *Emotional memory – recognition* | 2.63 (0.55) | – |
| *Emotional memory – priming* | 0.41 (0.09) | – |
| *Emotion regulation – reactivity* | 5.29 (1.63) | – |
| *Emotion regulation – reappraisal* | 0.038 (1.11) | – |
| *Hotel-task* | 301.93 (172.36) | – |
| *PSQI score (median, range)* | 4 (0–22) | – |
| ***HADS score (median, range)*** | 5 (0–20) | – |

in connectivity and imaging acquisition-related noise, since it is less affected by, for example, region of interest (ROI) inaccuracies (*Haak et al., 2018*). The aims of our study were to: (1) identify the functional gradients of the LC and reproduce them across different cohorts and image resolutions, using a 3T (the Cambridge Centre for Ageing and Neuroscience or CamCAN cohort) and a 7T (the Human Brain Connectome or HCP 7T cohort) datasets; (2) characterize how the functional gradients of the LC change over the course of aging in the CamCAN dataset, a large cohort of 618 individuals with resting-state fMRI between 18 and 88 years of age; (3) identify LC projections to cortical and subcortical areas; and (4) detect relevant associations between LC gradients with cognitive, emotional/affective, and physiological functions that decline with increasing age.

We show that the LC exhibits a basic rostro-caudal functional gradient that is reproducible across different cohorts and age ranges. The LC functional gradient is driven by different patterns of connections to cortical and subcortical areas that are relevant to understand the pathophysiological mechanisms of neurodegenerative disorders. Moreover, we demonstrate that individual differences in the finer spatial layout of the rostro-caudal gradient are related to age, and functions ascribed to the noradrenergic system correlate with gradient spatial characteristics when corrected for the effects of age. Our findings provide an alternative model of LC functional organization that might explain why distinct functional alterations of the LC could lead to different cognitive and psychiatric disorders.

## Results
### Functional connectivity is organized along a rostro-caudal axis in the LC

Characteristics of both cohorts can be found in *Table 1*. The average dominant gradient consistently exhibited a rostro-caudal organization in all age groups in the CamCAN dataset and across all subjects in the HCP 7T dataset (see *Figure 1A*; also see *Figure 1—figure supplement 1* for a difference map of gradients from the two datasets). To assess the extent to which the ROI mask influences the rostro-caudal gradient, we repeated the gradient calculation in both datasets using a larger LC ROI from *Tona et al., 2017*; gradients using this ROI retained the rostro-caudal organization (see *Figure 1—figure supplement 2*). The gradient was monotonous, meaning that the stepwise changes in connectivity

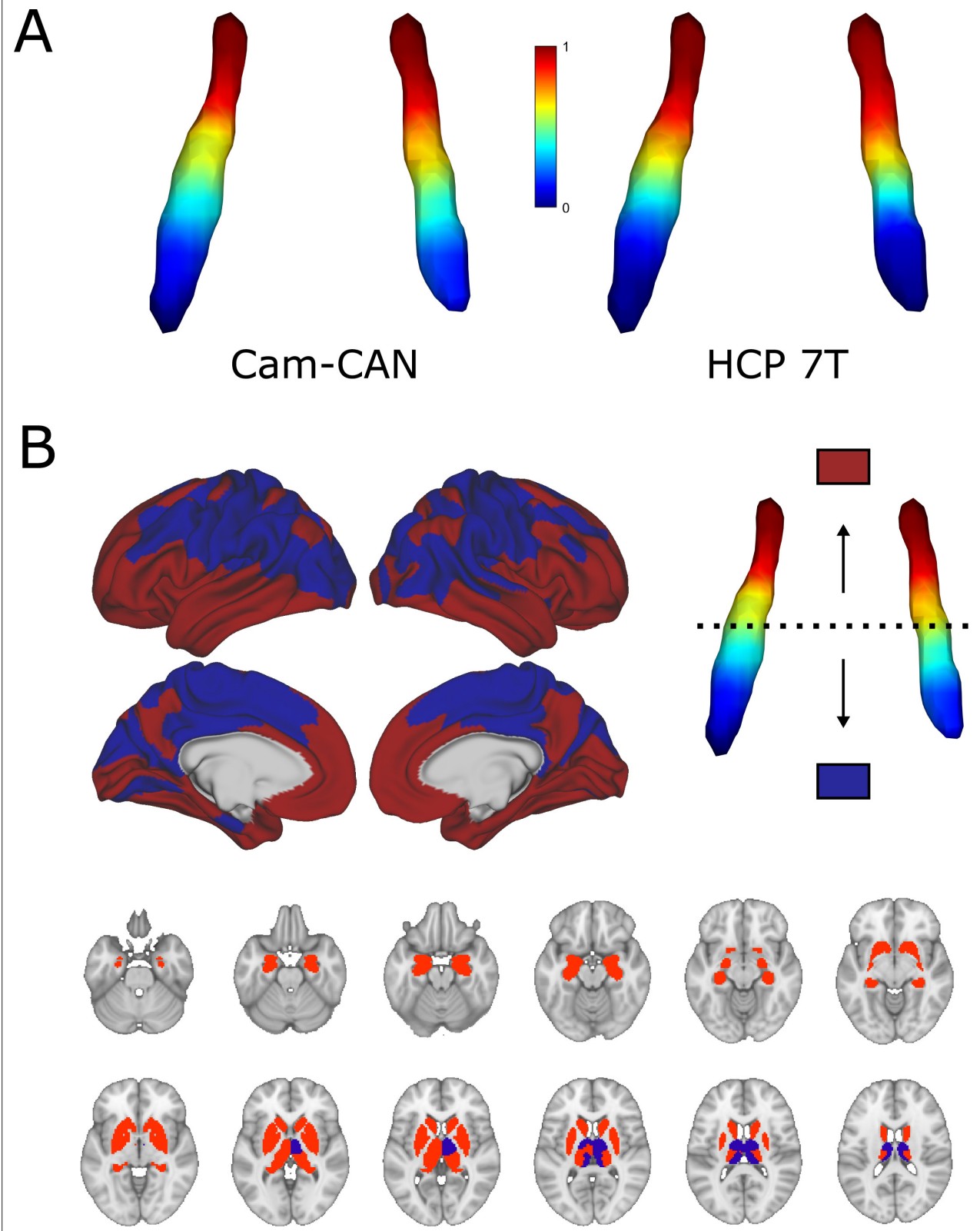

**Figure 1.** Average functional gradients of the locus coeruleus (LC) in the CamCAN 3T and Human Connectome Project (HCP) 7T datasets. (**A**) Average functional gradients from subjects below 40 years of age in the CamCAN and HCP 7T datasets were projected to a surface rendering of the LC region of interest (ROI) mask. We present younger participants from the CamCAN cohort to make it comparable to the demographic characteristics of the HCP cohort. The color bar denotes normalized values of the dominant eigenmap. (**B**) Projection maps of the LC were calculated as the mode of maximal

*Figure 1 continued on next page*

*Figure 1 continued*

correlation localization to either the rostral (red) or the caudal (blue) part of the LC in order to show the two extremes of connectivity. Projection maps were overlaid on the average HCP cortical surface and MNI152 template for the subcortical structures.

The online version of this article includes the following figure supplement(s) for figure 1:

**Figure supplement 1.** Normalized difference map depicting the similarity of the gradient between subjects below 40 years of age from the CamCAN and the Human Connectome Project (HCP) 7T dataset.

**Figure supplement 2.** Average functional gradients of the locus coeruleus (LC) in the CamCAN 3T and Human Connectome Project (HCP) 7T datasets using an LC region of interest (ROI) different from the main analysis.

were consistent along the longitudinal axis of the LC and did not show clear functional area boundaries. When partitioned into rostral and caudal parts, the rostral part was characterized by connections to the posterior and anterior cingulate cortices and the medial temporal lobe, whereas the caudal part projected to the parietal and visual cortices. Regarding subcortical structures, the rostral LC mainly projected to the hippocampi, amygdalae, and striatum, while the caudal part projected to the thalami, occipital, precentral, and postcentral regions (see *Figure 1B*). To summarize, the rostral part was mainly associated with higher order, associative cortical areas and limbic regions, while the caudal part was more connected to primary sensorimotor and visual areas.

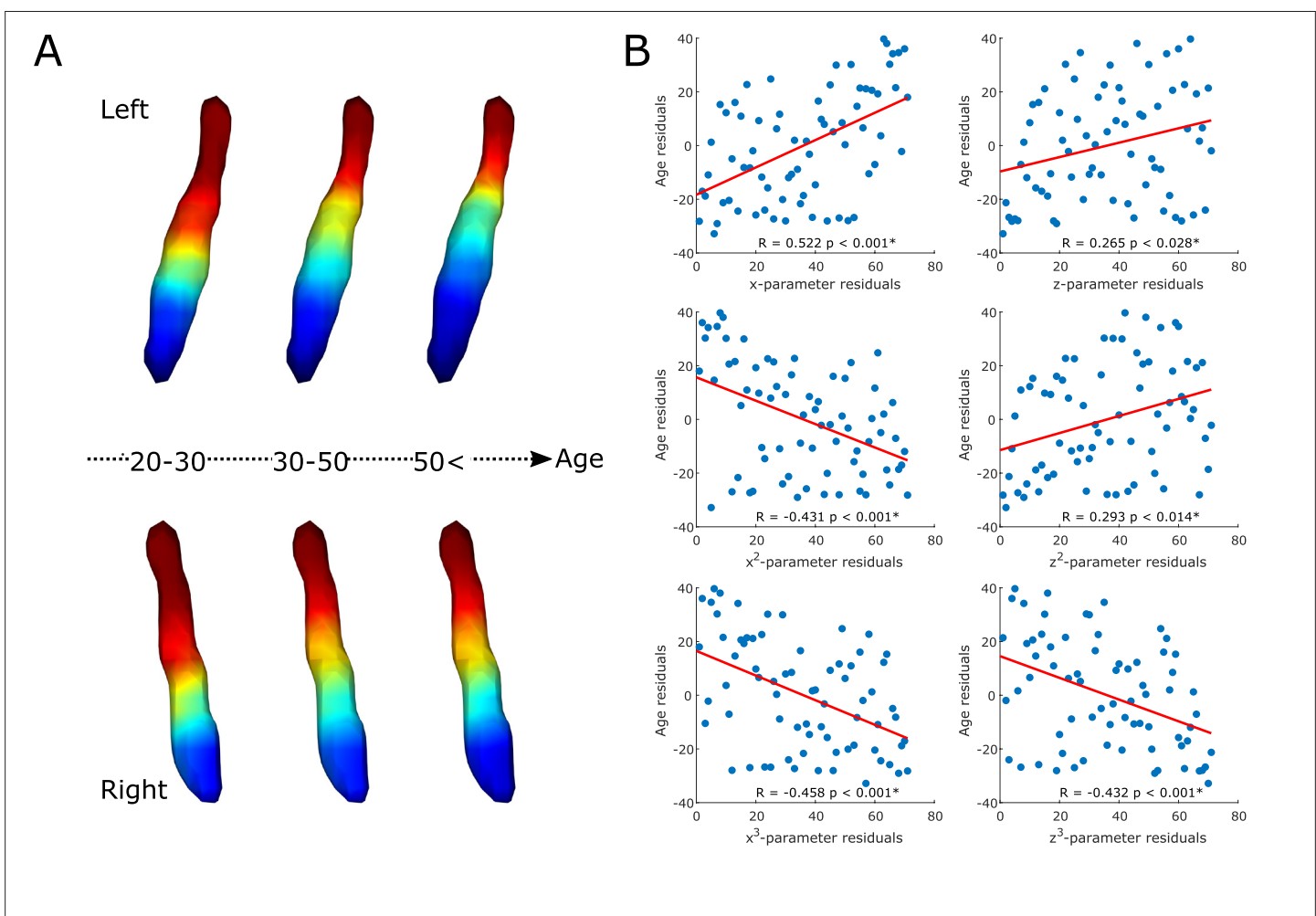

**Figure 2.** Age-related changes in the locus coeruleus (LC) functional gradient. Average functional gradients from representative age groups (20–30, 30–50, <50) were projected to a surface rendering of the left and right LC region of interest (ROI) mask (**A**). The scatter plots show the relationship between age and rank-transformed values of trend surface model (TSM) parameters (after adjusting for effects of sex) describing the spatial features of the gradient (see Materials and methods for more details) with a least-squares line in red (**B**). Parameter changes correspond to a loss of rostral-like connectivity, increased asymmetry, and more clustered functional organization of the LC in older participants.

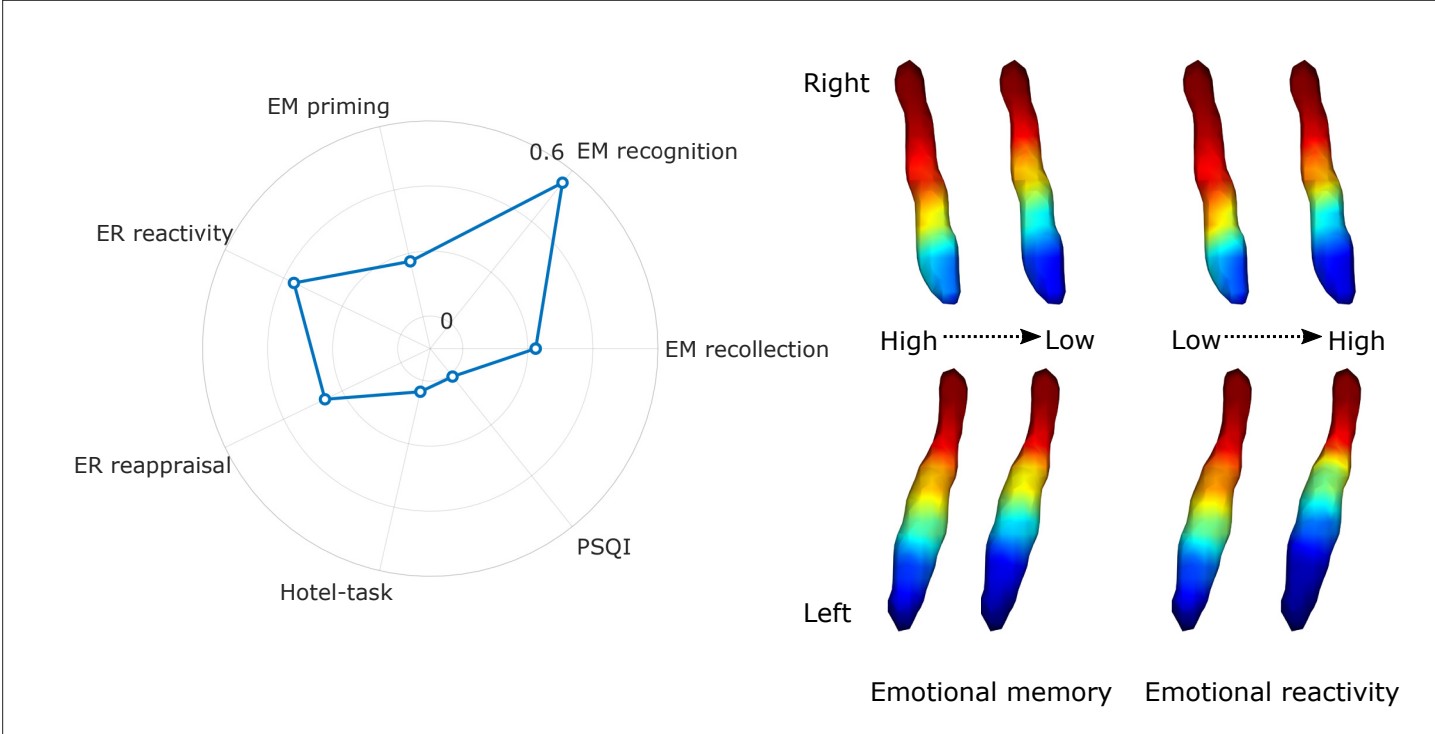

**Figure 3.** Association of locus coeruleus (LC) functional gradient spatial features and behavioral measures. The polar plot depicts additional variance explained by spatial features of the LC functional gradient over and above nuisance variables (age, sex, education for the analyses to predict cognitive variables) in emotional memory, emotion regulation, executive function, and sleep quality in terms of adjusted partial $R^2$ values. Example window-averaged gradients from participants with lowest and highest emotional memory scores and emotion regulation scores were projected to a surface rendering of the LC region of interest (ROI) mask, showing that old-like functional gradient spatial features (less extensive rostral-like connectivity and more clustered functional organization) are associated with lower scores in emotional memory and emotion regulation related to stimuli with negative valence. No significant associations were found with executive function and sleep quality. Abbreviations: EM = emotional memory, ER = emotion regulation, PSQI = Pittsburgh Sleep Quality Index.

The online version of this article includes the following figure supplement(s) for figure 3:

**Figure supplement 1.** Correlation plots of individual trend surface model parameters and cognitive scores.

## The functional gradient in the LC is different across ages

The gradient explained a significant amount of variance in age over and above the effects of sex (F=8.671, adjusted partial $R^2$=0.504, p<0.001; see *Figure 2A and B*). Also, specific parameters of the gradient's spatial model were associated with age in a way that a loss of rostral-like connectivity, more clustered appearance, and increased asymmetry were related with older age (see correlation plots on *Figure 2B*). This was further confirmed by the clusterability analysis, which showed that, with increasing age, the gradient could be clustered into rostral and caudal regions more effectively (R=0.313, p<0.009).

## Spatial features of the LC functional gradient are associated with behavioral measures

Regarding cognitive variables, the gradient explained a significant amount of variance above the effects of age, sex, and education in emotional memory scores – more specifically recollection of stimulus valence (F=2.632, adjusted partial $R^2$=0.224, p=0.016), recognition of previously seen objects associated with emotionally charged stimuli (F=7.992, adjusted partial $R^2$=0.552, p<0.001), priming (F=2.198, adjusted partial $R^2$=0.175, p=0.042), emotional reactivity to negative events (F=4.008, adjusted partial $R^2$=0.365, p<0.001), and reappraisal of emotional valence for negative stimuli (F=2.827, adjusted partial $R^2$=0.259, p=0.012) after false discovery rate (FDR) corrections. For a depiction of gradient spatial layouts associated with behavioral scores, see *Figure 3*. Beside the overall spatial features, some of these tests were also associated with specific parameters of the spatial

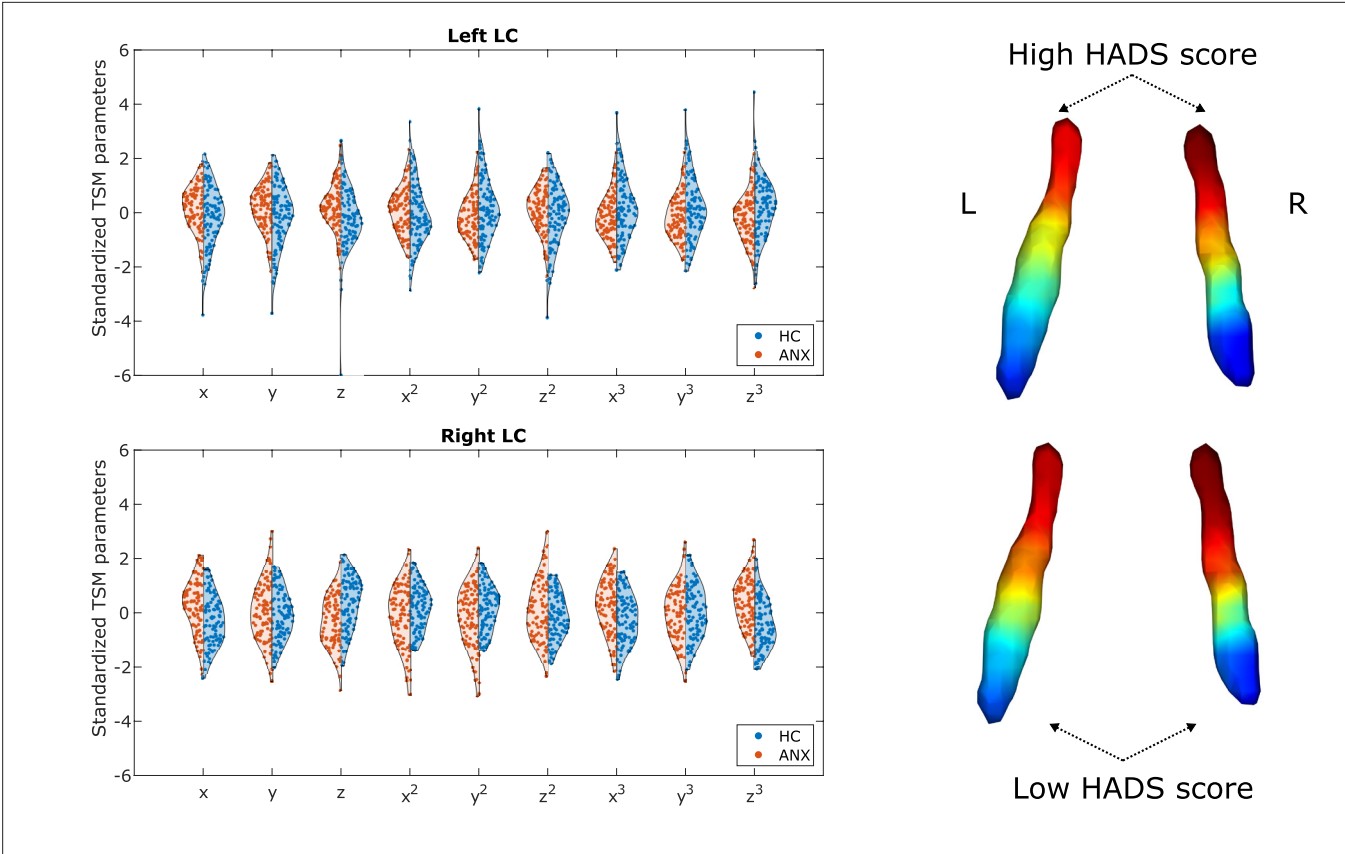

**Figure 4.** Alterations of the locus coeruleus (LC) functional gradient in participants with abnormal Hospital Anxiety and Depression Scale (HADS) scores. The violin plots depict gradient spatial model parameter distributions in the two groups. Example gradients from participants with low and high HADS scores were projected to a surface rendering of the LC region of interest (ROI) mask, showing that high HADS scores were associated with a reduction of rostral-like connectivity, as well as a greater asymmetry in the spatial layout of the bilateral LC gradients.

model, indicating a spatial gradient phenotype that is associated with lower scores of emotional memory and emotion regulation regardless of age, sex, or education (see *Figure 3—figure supplement 1* for partial Spearman's rank correlation coefficients and statistical significance). In general, an old-like functional gradient exhibiting a loss of rostral-like connectivity and more clustered functional organization was associated with worse performance in emotional memory and emotion regulation tasks related to negative events. Overall spatial features of LC functional gradients did not explain a significant amount of additional variance in executive functions (Hotel-task; F=1.473, adjusted partial $R^2$=0.036, p<0.168) or sleep quality scores (Pittsburgh Sleep Quality Index [PSQI]; F=1.124, adjusted partial $R^2$=0.01, p<0.352).

### The LC functional gradient is different in people with higher anxiety and depression ratings

Participants with higher-than-normal HADS scores exhibited a different gradient compared to participants with normal HADS scores (one-way multivariate analysis of variance [MANOVA]: p<0.001). The gradient in the pathological group showed generally less rostral-like connectivity and was more asymmetric (left LC – z: p<0.002, $z^3$: p<0.007; right LC – x: p<0.001, z: p<0.001, $x^3$: p<0.001, $z^3$: p<0.001; see *Figure 4*), showing a further reduction of left rostral-like connectivity compared to the right one.

### Discussion

In this study, we show that functional connectivity in the LC is organized along a rostro-caudal gradient. This functional gradient was reproduced in two independent cohorts, and was driven by differential connections to the medial temporal lobe, hippocampus, amygdalae, striatum, medial parietal cortex,

and anterior cingulate (from the rostral part) and the visual, sensorimotor cortices and thalamus (from the caudal part). Importantly, we show that spatial variation and rostral-caudal balance showed age-related changes: specifically, a loss of rostral-like connectivity, a more clustered functional organization over the entire LC, and an increased asymmetry were associated with older age. Moreover, these differences in the spatial characteristics of the LC gradient were associated with poorer emotional memory, emotion regulation, higher anxiety, as well as depression scores, independently of age, sex, and education. These findings highlight the essential role played by the LC in regulating memory, emotion, and psychiatric symptoms through its heterogenous patterns of functional connections projecting to different cortical and subcortical brain areas.

A rostral-caudal division has already been reported in the LC by studies using structural imaging and LC-contrast MRI sequences (*Dahl et al., 2019*). This division seems to be driven by a heterogeneous distribution of cellular, molecular, and connectivity-related characteristics along the LC. Specifically, LC-contrast MRI sequences such as magnetization transfer imaging seem to be associated with neuronal integrity in the LC, although the biological sources of this contrast are still not fully understood (*Keren et al., 2015*). A post-mortem study showed that noradrenergic neural somas in the LC show a specific distribution characterized by a dense collection of cell bodies in the central LC that disperses in both caudal and rostral directions, with a lower number of these noradrenergic cells in the caudal part compared to the rostral part (*Fernandes et al., 2012*). Using a gradient-based approach applied to functional imaging, we reproduced this structural division, revealing for the first time a rostro-caudal functional organization consistent across cohorts with different acquisition parameters.

In addition to distinct cellular distributions, spatial heterogeneity in the LC is also driven by projections to different brain areas along its longitudinal axis. Tracer studies show that the rostral-dorsal part of the LC projects to neocortical regions, the hippocampus, medial temporal lobe, and amygdala, whereas the caudal-ventral part sends its main efferent projections to the cerebellum and medulla (*Samuels and Szabadi, 2008b*). Here, we provide an in vivo account of LC functional projections, revealing that the connections most reproducible across cohorts, mainly from the rostral part, are in reasonable agreement with anatomical connectivity studies, and these connections drive the gradient of functional connectivity throughout the LC. However, in addition to tracer studies, we found that the caudal part also shows connectivity to primary sensorimotor and visual cortical regions, as well as the thalami. Some of these regions, especially the thalami, were reported to show connectivity with the LC in previous fMRI studies (*Liebe et al., 2020*). Regarding primary sensorimotor and visual regions, it is possible that downstream multisynaptic connections in these regions reflect the cerebellar projections of the caudal LC, since the cerebellum is known to have strong reciprocal connections to multiple thalamic nuclei and primary sensorimotor cortical areas (*Caligiore et al., 2017*). It is important to note that the fMRI acquisition in the CamCAN cohort did not cover the entire cerebellum and upper cervical spinal cord, which are known targets of projections in the caudal LC, therefore we did not investigate them in this study.

Since in vivo investigation of the LC is hindered by its small size and location, the functional implications of the heterogeneous structure and connectivity of the LC have not been fully explored. Electrophysiological studies report non-overlapping spontaneous activation dynamics from different sites in the LC that associate with and drive different cortical states, suggesting the presence of distinct neural ensembles in the LC that regulate multiple cortical states, instead of a homogeneous arousal-related release of noradrenaline across the cortex (*Noei et al., 2022*). Although fMRI provides a comprehensive view of LC function due to this nucleus' extensive and diffuse cortical projections, this technique is limited by low spatial resolution, which could explain why previous resting-state fMRI studies focused on the LC obtained inconsistent results (*Liebe et al., 2020*; *Song et al., 2021*). The high connectivity heterogeneity of the LC shown in our study indicates that using the LC as a single, homogeneous ROI might not be the best approach to assess connectivity in this region. We show that a gradient-based approach is able to uncover the functional organization of the LC, and that it is relevant to LC-related behavioral measures. Specifically, we found that the LC functional connectivity gradient shows age-related differences, and its spatial features correlated with several behavioral measures that were previously linked to the noradrenergic system. Functional connectivity of the LC has been demonstrated to depend on age, especially connections to the medial temporal lobe and amygdala projecting from the rostral part of this region (*Song et al., 2021*). In line with this, we observed a loss of rostral-like connectivity with increasing age. Other in vivo MRI studies also reported

that most age-related structural changes seem to be confined to the rostral part of the LC (*Liu et al., 2020*; *Betts et al., 2017*; *Liu et al., 2019*), and can be linked to age-related alterations in cortical thickness (*Bachman et al., 2021*). These changes possibly stem from morphological alterations that affect neuromelanin-containing neurons in the rostral part, leading to a reduction in cell size and potentially cell loss, which might be associated with alterations in their projections as well (*Chan-Palay and Asan, 1989*). Another possible explanation is that, with advancing age, misfolded protein aggregates build up in the brain that may or may not cause manifest neurodegeneration later on *Farrell et al., 2022*. Recent research identified the LC as an early site for tau deposition, where accumulation of pathological proteins increases the risk for neurodegenerative conditions like AD (*Jacobs et al., 2021*). Recent histological studies reported that the distribution of pathological proteins (especially tau neurofibrillary tangles) is non-uniform in the LC, preferentially affecting the rostral-dorsal part, irrespective of Braak stages (*Gilvesy et al., 2022*). Early morphological changes in tau-containing LC neurons included gradual dendritic atrophy and changes in axonal morphology indicative of impaired axonal transport, which might contribute to earlier loss of connectivity in the rostral LC section during aging. Our results support these age-related effects on LC integrity and provide a functional imaging marker to assess accompanying regional changes in LC connectivity in vivo.

Independently of age, an old-like functional gradient was also associated with worse performance on emotional memory and emotion regulation scores, consistent with previous studies showing that lower rostral LC integrity is associated with worse memory performance in general (*Dahl et al., 2019*). A possible explanation for this is the relationship between pathological amyloid-β buildup and decreased functional connectivity of the LC to the hippocampus and amygdala during the encoding of salient events (*Prokopiou et al., 2022*). These connections were more characteristic of the rostral part of the LC in our study, where poorer memory performance was associated with a loss of rostral-like connectivity. Other studies further detail the association between LC integrity and memory, showing that the link between LC integrity and memory performance is more emphasized when participants try to recall aversive effects, which reflects the role of the noradrenergic system in the formation of emotionally charged memories (*Hämmerer et al., 2018*). Furthermore, the LC functional gradient was associated with emotion regulation, a process that is strongly associated with fear learning and development of anxiety, which have been linked to the noradrenergic system (*Uematsu et al., 2017*). The fact that the LC functional gradient reflected emotional memory performance and emotion regulation in our study independently of age points to the selectivity of the LC gradient in explaining these behavioral measures and its viability as a marker for emotional processing. Interestingly, we did not find an association between the LC functional gradient and executive functions or sleep quality. Although more detailed studies are needed, it is possible that the association between executive function and the LC is mostly mediated by age. Regarding the PSQI sleep quality indices, this questionnaire is a subjective assessment of sleep quality and more objective studies are needed to examine the effects of LC functional gradients on sleep architecture.

Furthermore, participants with higher anxiety and depression ratings exhibited a more asymmetric functional gradient, with the same parameters affected as in our results concerning aging. This is in line with previous studies reporting a role for the LC in the generation of anxiety and depression (*Morris et al., 2020a*). The LC modulates stress-related arousal through its ascending norepinephrinergic projections, and chronic stress involved in the pathogenesis of clinical anxiety amplifies tonic activity in the LC in response to subsequent stressors in animal models (*Jedema et al., 2008*). Additionally, chronic stress is associated with differences in dendritic morphology in the prefrontal cortex of rats, which might impact efferent projections from the LC (*Liston et al., 2006*). Anxiety symptoms also seem to come with degenerative or maladaptive structural changes within the LC, since this condition was associated with morphological differences of the LC on ultra-high field MRI in animal (*Omoluabi et al., 2021*) and human studies (*Morris et al., 2020b*). Although the relevance of the LC in depression is mostly tied to its role in the development of anxiety and therefore stress-induced depression (*Weiss et al., 1994*), there are several studies that report specific alterations of the LC in for example major depression disorder (MDD). A task fMRI study found decreased LC functional connectivity during a visual oddball task in patients with MDD (*Del Cerro et al., 2020*). Apart from functional alterations, the LC also shows reduced structural integrity in patients with MDD (*Guinea-Izquierdo et al., 2021*), as well as molecular changes that manifest in lower levels of neuronal nitric oxide synthase, an intracellular mediator of glutamatergic signaling, in the LC (*Karolewicz et al., 2004*). Regarding

lateralized alterations of the LC functional gradient in participants with elevated anxiety and depression scores, asymmetry can be physiological to a point, since a lateralized signal distribution of the LC was observed in healthy adults in previous studies (*Betts et al., 2017*). Although future studies are needed to confirm this, it is possible that a more asymmetric LC functional gradient accompanying higher HADS scores reflects lateralized amygdala response alterations reported in previous studies of patients with anxiety disorders with accompanying plastic changes in LC connectivity (*Evans et al., 2008*). However, it is important to note that the HADS score represents a general assessment of a person's anxiety and depression levels, and more specific and detailed studies will be needed to elucidate how LC functional gradients are associated with anxiety and depressive symptoms.

Interestingly, we observed lateralized changes in the LC gradients both in connection with aging and cognitive performance. Generally, the LC connects to several highly lateralized cortical networks, for example, the salience and frontoparietal attention networks, that might result in an asymmetric plasticity in the LC. Neurodegenerative disorders seem to affect the left LC more both in terms of integrity and connectivity. For example, more widespread loss of connectivity between the left LC and resting-state networks was found in PD patients, with a correlation between left LC-executive control network connectivity and cognition (*Sun et al., 2023*). Another study found decreased connectivity between the left LC and the parahippocampal gyrus in mild cognitive impairment (*Jacobs et al., 2015*). Furthermore, increased left LC contrast was associated with aging and PD as well (*Ye et al., 2022*; *Betts et al., 2017*). However, the biological basis for this is elusive, as post-mortem studies generally find the bilateral LC symmetric and mostly report pathological changes in the rostral and middle LC (*Beardmore et al., 2021*). In our case, a possible interpretation is that with the loss of rostral-like connectivity, previously rostral-like areas lose their specific connections and become more similar to the caudal part in terms of connectivity. In this study, since we did not investigate the cerebellum and the spinal cord, the typical caudal connectivity profile is more non-specific, since some of its dominant connections are not assessed.

This study has some limitations. Although we reproduced the functional gradients in a dataset with higher spatial resolution of 7T, the spatial resolution and coverage of 3T functional MRI scans is a limiting factor warranting further studies, as well as the fact that physiological data (such as pupil size tracking or respiratory monitoring) was not acquired during the 3T scans. A further technical limitation is that connectopic mapping requires a measure of spatial smoothness for gradient calculation, therefore we used a smoothing kernel of 3 mm FWHM and constrained it to the LC mask in order to avoid signal leakage. Also, the dataset used to investigate age-related changes is cross-sectional, and longitudinal studies need to be conducted in order to confirm the age-related differences we observed here. Because of the small ROI size and the limited temporal resolution of the 3T dataset, we employed a group-based approach to calculate the gradients, however, this limits the assessment of gradient reproducibility on the individual level, which should be explored in further studies. Furthermore, although the behavioral significance of functional gradients is increasingly recognized in correlational studies, further research is required to obtain more information about their underlying physiological role.

In conclusion, in this study we show that functional connectivity is organized into a gradient along the longitudinal axis of the LC. This functional gradient was reproducible across cohorts and different fMRI protocols, and showed relevant associations with age, memory, emotion, as well as anxiety and depression ratings. Therefore, our study provides an in vivo account of functional heterogeneity and organization in the LC, as well as a possible imaging marker for LC-related behavioral measures of memory and psychiatric symptoms.

## Materials and methods
### Participants
We included 618 participants who had resting-state fMRI data from the CamCAN dataset (https://www.cam-can.org). To investigate the relationship between LC functional gradients and cognitive measures, we assessed a set of cognitive functions that were previously associated with LC function (*Liu et al., 2020*), namely negative valence trials of emotional memory and emotion regulation tests (*Hämmerer et al., 2018*), executive functions measured by the Hotel-task (*von der Gablentz et al., 2015*), sleep quality measured by the PSQI (*Buysse et al., 1989*), as well as anxiety and depression

symptoms measured by the HADS (*Zigmond and Snaith, 1983*). From the imaging dataset, 292 participants had emotional memory scores, 272 emotion regulation scores, 601 Hotel-task scores, 586 PSQI scores, and 615 had HADS scores. Demographic characteristics of these subgroups did not differ from that of the whole sample.

To investigate the reproducibility of the functional gradient results, we analyzed 184 participants from the HCP with 7T resting-state fMRI data (https://db.humanconnectome.org). The first resting-state fMRI run was analyzed for all subjects.

## Image acquisition

The 3T resting-state fMRI protocol for the CamCAN dataset included an echo planar imaging (EPI) sequence (TR = 1970 ms; TE = 30 ms; FOV: 192×192 mm$^2$; 3×3×4.4 mm$^3$ voxel resolution; duration: 8.5 min); data was collected at a single site using a 3T Siemens TIM Trio scanner with a 32-channel head coil (*Taylor et al., 2017*). Participants were asked to lie motionless with their eyes closed and not fall asleep during the scan.

The acquisition protocol for the HCP 7T resting-state fMRI data included a multiband gradient-echo EPI sequence (TR = 1000 ms; TE = 22.2 ms; FOV: 208×208 mm$^2$; 1.6 mm isotropic voxel resolution; duration: 16 min); data was collected at a single site using a 7T Siemens Magnetom scanner (*T Vu et al., 2017*). During the acquisition, participants were asked to keep their eyes open and fixate on a cross, shown on a presentation screen inside the scanner.

## Image preprocessing

The HCP 7T dataset was preprocessed according to a minimal preprocessing pipeline as described in *Glasser et al., 2013*. First, fMRI scans underwent EPI distortion correction, removal of non-brain tissue, motion correction, intensity normalization, and a two-stage registration to standard MNI space, followed by regression of noise and motion-related artifacts using FSL FIX (https://fsl.fmrib.ox.ac.uk/fsl/fslwiki/FIX) and high-pass filtering with a cut-off of 2000s. Additionally, following previous publications using functional gradients (*Przeździk et al., 2019*; *Haak et al., 2018*), signals from the cerebrospinal fluid and white matter were removed via nuisance regression.

Functional scans from the CamCAN dataset were preprocessed through a standard pipeline implemented in fMRIPrep (v20.2.4). After removing the first two volumes to allow for steady-state magnetization, functional images were corrected for motion and slice timing effects, skull-stripped and co-registered to a standard 2 mm resolution MNI152 template space using a two-stage registration approach with Freesurfer (*Dale et al., 1999*) and ANTs (*Avants et al., 2014*). Nuisance regression was performed to further correct for motion effects (employing the 24-parameter head motion model; *Friston et al., 1996*) and to remove confounding signals from the cerebrospinal fluid and white matter.

To minimize signal leakage from the surrounding brainstem regions and the fourth ventricle, the LC was masked for each subject and spatial smoothing was applied with a 3 mm FWHM isotropic 3D Gaussian kernel within the mask to ensure that weighted graphs obtained from the similarity matrices are fully connected, since a measure of spatial smoothness is necessary for gradient calculation (*Haak et al., 2018*; see Connectopic mapping section).

## Delineation of the LC

The LC was delineated using a standard mask from a recent study that developed a meta-analytical consensus mask by aggregating previous segmentations of the LC on neuromelanin-sensitive MRI sequences (*Dahl et al., 2022*). The mask was transformed to standard MNI152 space and downsampled to the respective isotropic spatial resolution of the two preprocessed datasets (2 mm isotropic voxels in the CamCAN dataset and 1.6 mm isotropic voxels in the HCP 7T dataset).

## Connectopic mapping

After preprocessing, we performed connectopic mapping, a data-driven approach that characterizes spatially continuous changes in connectivity profiles as gradients (*Haak et al., 2018*), using the Conn-Grads toolbox (https://github.com/koenhaak/congrads; *Haak et al., 2018*). The analysis starts by calculating functional connectivity matrices (or 'fingerprints') between all voxel time series in an ROI (in this case this was the LC defined in the way described above), and time series from a target mask. In the current study, the target mask was defined using a cortical and subcortical parcellation. Average time

series were extracted from each parcel in the atlases separately, and then pairwise correlations were calculated with time series from all voxels in the ROI (the LC). We used the Glasser-atlas (containing 360 parcels) as cortical and the Tian-atlas (containing 50 parcels) as subcortical parcellation (*Glasser et al., 2016*; *Tian et al., 2020*). Afterward, we obtained a similarity matrix from the functional connectivity matrices of LC ROI voxels by calculating the eta-squared measure (*Cohen et al., 2008*). The similarity matrix contains the eta-squared similarity measure of fingerprints between all pairs of LC voxels, therefore one index in the similarity matrix corresponds to the similarity between the fingerprints of two specific LC voxels. This similarity matrix was then transformed into a connected graph and fed to the Laplacian eigenmaps algorithm (*Belkin and Niyogi, 2003*), which results in a decomposition of the similarity graph so that the eigenvectors of graph Laplacian represent axes along which the stepwise change in functional connectivity is the largest, that is spatially graded patterns of functional organization, or functional gradients. Along these gradients, voxels which exhibit similar functional connectivity fingerprints will be placed close to each other. The spatial features of these gradients were then summarized using a spatial regression approach (trend surface modeling) to improve sensitivity over voxel-wise analyses. Trend surface modeling uses polynomial basis functions to describe the spatial features of a functional gradient (*Huertas et al., 2017*). We fit a third-order trend surface model (TSM) based on previous studies (*Przeździk et al., 2019*; *Oldehinkel et al., 2022*; *Marquand et al., 2017*) that contains nine parameters corresponding to the three axes of MNI152 space using Bayesian linear regression (coefficients for x, y, z axes and their second and third power: $x^2$, $y^2$, $z^2$, and $x^3$, $y^3$, $z^3$). More generally speaking, first-order coefficients of the TSM describe the slope of changes in functional connectivity along the x, y, z axes, while higher order coefficients describe more detailed spatial features (such as curvature).

## Statistical analysis

Statistical analysis was performed using RStudio (v2022.07.1, with the *rsq* package, v2.5: https://cran.r-project.org/package=rsq) and MATLAB (MathWorks Inc, R2021B). Since individual gradient estimation is often not consistent (*Haak et al., 2018*), we derived a group-based scheme to estimate the relationship between behavioral variables and LC gradient parameters.

To estimate the relationship between the gradients and age, we estimated average gradients for each year (from 18 to 88 years) and calculated their association with increasing age using Spearman's partial rank correlation coefficients.

To assess how continuous cognitive scores are related to the gradients, we employed a sliding window correlation approach. More specifically, participants were sorted into ascending order based on the cognitive test. Then, an average gradient and cognitive score, as well as nuisance variables were calculated for a window of subjects. Here, we chose a window length of 20 subjects to preserve statistical power for the correlations while also making sure that the gradient estimation was consistent. The window was then slid in steps of five subjects, and the procedure was repeated for all windows. The resulting window-averaged values were then included in a linear model and F-tests were used to compare the linear model that included the gradient spatial model parameters and nuisance variables to a null model containing only nuisance variables, similarly to previous studies that used connectopic mapping (*Przeździk et al., 2019*). The partial coefficient of determination from the F-tests was calculated to obtain explained variance of the first model over and above the effects of nuisance variables. Additionally, Spearman's partial rank correlation was calculated between the windowed gradient parameters and cognitive scores, controlling for the effects of age, sex, and education to see if parameters of the spatial model could be used as specific markers of cognitive performance. For the F-tests and post hoc correlations, FDR corrections were applied to correct for multiple comparisons.

Furthermore, to check how the clusterability of the gradient (i.e., how clearly the rostral and caudal region differs in terms of connectivity), we carried out the following analysis based on *Ngo et al., 2021*. Gradient maps were clustered into k=2 clusters using the k-means clustering algorithm, then the Calinski-Harabasz criterion was calculated for each individual gradient map, which was then used as a measure of clusterability and correlated with age by calculating the partial Spearman's rank correlation coefficient controlling for the effects of sex.

The HADS was used to stratify the participants into people with and without depressive and anxiety symptoms (HADS anxiety and/or depression scores below and above 7; *Zigmond and Snaith, 1983*).

Since the resulting groups were highly imbalanced (499 participants with normal, 116 with abnormal HADS scores), we performed bootstrapping to obtain an equal sized sample of group average gradients (drawing 20 subjects for each iteration) from the two groups. Spatial model parameters were compared between the two groups using a one-way MANOVA. Afterward, post hoc Mann-Whitney U-tests were carried out for each spatial model parameter. Correction for multiple comparisons was performed via FDR.

## Acknowledgements

DV and JBP are supported by the Swedish Research Council, Alzheimerfonden, Brain Foundation, Dementia Foundation, Karolinska Institute, Stratneuro, Center for Medical Innovation, Stohnes and Gamla Tjänarinnor. HZ is a Wallenberg Scholar supported by grants from the Swedish Research Council (#2022-01018), the European Union's Horizon Europe research and innovation programme under grant agreement No 101053962, Swedish State Support for Clinical Research (#ALFGBG-71320), the Alzheimer Drug Discovery Foundation (ADDF), USA (#201809-2016862), the AD Strategic Fund and the Alzheimer's Association (#ADSF-21-831376C, #ADSF-21-831381C, and #ADSF-21-831377C), the Bluefield Project, the Olav Thon Foundation, the Erling-Persson Family Foundation, Stiftelsen för Gamla Tjänarinnor, Hjärnfonden, Sweden (#FO2022-0270), the European Union's Horizon 2020 research and innovation programme under the Marie Skłodowska-Curie grant agreement No 860197 (MIRIADE), the European Union Joint Programme – Neurodegenerative Disease Research (JPND2021-00694), and the UK Dementia Research Institute at UCL (UKDRI-1003). HLJ is supported by National Institutes of Health (R01AG062559, R01AG068062, R21AG074220), Alzheimer's Association AARG-22-920434 and Alzheimer Nederland WE.03-2019-02. MJB is supported by the Deutsche Forschungsgemeinschaft (DFG, German Research Foundation) – Project-ID 425899996 – SFB 1436 and CBBS NeuroNetzwerk 17 and by the German Federal Ministry of Education and Research (BMBF, funding code 01ED2102B) under the aegis of JPND.

## Additional information

### Competing interests

Henrik Zetterberg: Henrik Zetterberg has served at scientific advisory boards and/or as a consultant for Abbvie, Acumen, Alector, Alzinova, ALZPath, Annexon, Apellis, Artery Therapeutics, AZTherapies, CogRx, Denali, Eisai, Nervgen, Novo Nordisk, Optoceutics, Passage Bio, Pinteon Therapeutics, Prothena, Red Abbey Labs, reMYND, Roche, Samumed, Siemens Healthineers, Triplet Therapeutics, and Wave, has given lectures in symposia sponsored by Cellectricon, Fujirebio, Alzecure, Biogen, and Roche, and is a co-founder of Brain Biomarker Solutions in Gothenburg AB (BBS), which is a part of the GU Ventures Incubator Program (outside submitted work). The other authors declare that no competing interests exist.

### Funding

| Funder | Grant reference number | Author |
| --- | --- | --- |
| Swedish Research Council | | Joana B Pereira<br>Dániel Veréb |
| Alzheimerfonden | | Joana B Pereira |
| Brain Foundation | | Joana B Pereira<br>Dániel Veréb |
| Dementia Foundation | | Dániel Veréb<br>Joana B Pereira |
| Stohnes Stiftelse | | Joana B Pereira<br>Dániel Veréb |
| Gamla Tjänarinnor | | Dániel Veréb<br>Henrik Zetterberg<br>Joana B Pereira |

| Funder | Grant reference number | Author |
|---|---|---|
| Swedish Research Council | #2022-01018 | Henrik Zetterberg |
| HORIZON EUROPE European Research Council | 101053962 | Henrik Zetterberg |
| HORIZON EUROPE European Research Council | 860197 | Henrik Zetterberg |
| Swedish State Support for Clinical Research | #ALFGBG-71320 | Henrik Zetterberg |
| Alzheimer's Drug Discovery Foundation | #201809-2016862 | Henrik Zetterberg |
| Alzheimer's Disease Strategic Fund | #ADSF-21-831376-C | Henrik Zetterberg |
| Alzheimer's Disease Strategic Fund | #ADSF-21-831381-C | Henrik Zetterberg |
| Alzheimer's Disease Strategic Fund | #ADSF-21-831377-C | Henrik Zetterberg |
| Bluefield Project | | Henrik Zetterberg |
| Olav Thon Foundation | | Henrik Zetterberg |
| Erling-Persson Family Foundation | | Henrik Zetterberg |
| Hjärnfonden | #FO2022-0270 | Henrik Zetterberg |
| European Union Joint Programme - Neurodegenerative Disease Research | JPND2021-00694 | Henrik Zetterberg |
| UK Dementia Research Institute | UKDRI-1003 | Henrik Zetterberg |
| National Institutes of Health | R01AG062559 | Heidi IL Jacobs |
| National Institutes of Health | R01AG068062 | Heidi IL Jacobs |
| National Institutes of Health | R21AG074220 | Heidi IL Jacobs |
| Alzheimer's Association | AARG-22-920434 | Heidi IL Jacobs |
| Deutsche Forschungsgemeinschaft | 425899996 | Matthew Betts |
| Deutsche Forschungsgemeinschaft | SFB 1436 | Matthew Betts |
| Alzheimer Nederland | WE.03-2019-02 | Heidi IL Jacobs |
| CBBS NeuroNetzwerk 17 | | Matthew Betts |
| Bundesministerium für Bildung und Forschung | 01ED2102B | Matthew Betts |

The funders had no role in study design, data collection and interpretation, or the decision to submit the work for publication.

## Author contributions

Dániel Veréb, Conceptualization, Data curation, Formal analysis, Visualization, Methodology, Writing – original draft, Writing – review and editing; Mite Mijalkov, Anna Canal-Garcia, Yu-Wei Chang, Emiliano Gomez-Ruiz, Blanca Zufiria Gerboles, Giovanni Volpe, Matthew Betts, Heidi IL Jacobs, Methodology, Writing – review and editing; Miia Kivipelto, Per Svenningsson, Henrik Zetterberg, Writing – review

and editing; Joana B Pereira, Conceptualization, Methodology, Writing – original draft, Writing – review and editing

## Author ORCIDs
Dániel Veréb ⬤ https://orcid.org/0000-0003-2077-5252
Giovanni Volpe ⬤ https://orcid.org/0000-0001-5057-1846
Joana B Pereira ⬤ https://orcid.org/0000-0002-4604-2711

## Ethics
Human subjects: Regarding the HCP cohort, the imaging data used in this study was publicly available and anonymized. Participants provided their informed consent and the study was previously approved by the Washington University Institutional Review Board as part of the Human Connectome Project. The Cam-CAN cohort study was conducted in compliance with the Helsinki Declaration, and has been approved by Cambridgeshire 2 Research Ethics Committee (reference number: 10/H0308/50).

Reviewer #1 (Public Review): https://doi.org/10.7554/eLife.87188.3.sa1
Reviewer #2 (Public Review): https://doi.org/10.7554/eLife.87188.3.sa2
Author Response https://doi.org/10.7554/eLife.87188.3.sa3

# Additional files

## Supplementary files
• MDAR checklist

## Data availability
All imaging and behavioural data used in this study is openly available at https://camcan-archive.mrc-cbu.cam.ac.uk/dataaccess/ for the Cam-CAN cohort and at db.humanconnectome.org for the Human Connectome Project cohort.

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
