## [Editor Report · eLife assessment]

This study provides **fundamental** imaging evidence from two independent functional imaging datasets, for a rostral-caudal gradient of locus coeruleus connectivity, which changes across the lifespan. The gradient approach is well-established and **convincing** results were obtained and validated using large 3T and 7T fMRI datasets. This work will be of interest to clinical and cognitive neuroscientists.

---

## [Referee Report · Reviewer #1 (Public Review)]

This study provides valuable imaging evidence for the connectopic mapping of the locus coeruleus where a rostro-caudal gradient was linked to heterogeneous functional organisations of the structure. The functional gradient of the LC changes over ageing and reflects capacities of related brain functions. The gradient approach is well-established and solid results were obtained and validated using large 3T and 7T fMRI dataset. The work highlights the importance of using more specific spatial definition of the LC based on distinct connectivity patterns in future resting-state fMRI studies.

---

## [Referee Report · Reviewer #2 (Public Review)]

The authors have provided evidence for a rostral-caudal organisation of locus coeruleus connectivity, which they show (i) differs across the lifespan, (ii) is associated with relevant cognitive and mood measures. They have taken a data-driven, gradient-based approach, which was applied in the CamCan dataset and then replicated in the HCP dataset. This is a useful contribution to the field as it comprehensively shows a rostral-caudal pattern of connectivity in vivo, which has mostly been supported by tracer studies to date.

The strengths of the study are the large sample sizes and replication across two cohorts. The connectomic mapping approach they have applied is very well suited to the question at hand, as it allows a continuous gradient of organisation to be identified.

---

## [Author Response]

The following is the authors’ response to the original reviews.

**Reviewer #1 (Public Review):**
Comment 1. The authors used a meta-mask based on previous LC structural studies to delineate the LC on functional scans within two large public datasets (3T CamCAN and 7T HCP).The rostral part of the LC was characterized by connections to the posterior and anterior cingulate cortices, medial temporal lobe, hippocampus, amygdala and striatum, while the caudal part projected to the parietal cortex, occipital cortex, precentral and postcentral regions, and thalamus. Older ages were associated with less rostral-like connectivity and increased asymmetry. The gradient explained variance above the effects of age, sex and education on some emotional and cognitive measures. In particular, the old-like functional gradient (loss of rostral-like connectivity and more clustered functional organization) was associated with worse performance on emotional memory and emotion regulation tasks but not to executive functioning or self-rated sleep quality.Participants with higher anxiety and depression also showed less rostral-like connectivity and more asymmetry. Both the aging and the anxiety/depression asymmetry manifested as less rostral-like connectivity in the left LC than the right LC.A strength of this study is that it is the first to attempt a voxel-based approach to quantifying functional connectivity in the LC. The results finding differences between rostral and caudal LC connectivity patterns are broadly consistent with prior work indicating differences between rostral/caudal LC and should help advance understanding of the LC's connectivity patterns with cortical regions.

We thank the reviewer for the thorough and positive assessment of our manuscript.

Comment 2. A limitation of the study is the challenge of assessing activity not only from the small LC brainstem nucleus but also within it. Given the current spatial limitations of whole-brain functional imaging, the current findings are bolstered by including the 7T 1.6mm isotropic data. Spatial smoothing was applied with a 3mm FWHM isotropic kernel which may have reduced precision.

The reviewer raises valid points. Spatial resolution is indeed a limiting factor for assessing the LC with functional MRI. The choice of including spatial smoothing in the preprocessing was necessary because connectopic mapping requires a measure of spatial smoothness for the gradient calculation (see Haak et al. 2018 NeuroImage). We included a sentence explaining this in the revised version of the manuscript and added it also as an additional limitation.

Comment 3. Another limitation was that the authors made conclusions about clustered functional organization but it was not clear how clustering was quantified.

We thank the reviewer for the comment. Clusterability was quantified in the following way (based on Ngo et al. 2021 NeuroImage). First, gradient maps were clustered into k=2 clusters using the k-means clustering algorithm and then the Calinski-Harabasz criterion was calculated for each individual gradient map, which was used as a measure of clusterability. Higher criterion values were significantly associated with older age (Spearman’s rho = 0.3129, p<0.0089), indicating that the gradient was more clustered in older individuals. This new analysis is now included in the revised version of the manuscript.

**Reviewer #1 (Recommendations For The Authors):**
Comment 1. Would it be equally accurate to state that participants with higher anxiety and depression showed more caudal-like connectivity or are the differences clearly localized to the rostral LC?

We thank the reviewer for the question. Since the gradients are by default a dimensionless scale that was further normalized to the range of 0-1, both interpretations are possible. We hypothesized a loss of rostral-like LC connectivity based on previous literature.

Comment 2. These resting-state findings seem to show some interesting parallels to the structural rostral/caudal LC MRI contrast relationships with cortical thickness in Bachman et al. (2021, Neurobiology of Aging), who found that positive associations between LC contrast and structural thickness were found among older adults for rostral but not caudal LC (corresponding with the rostral regions showing the most age-related change). It is also interesting that in Bachman et al., younger adults showed negative correlations between caudal LC contrast and cortical thickness, which may relate to associations with a more caudal-like connectivity pattern (assuming this is a fair way to interpret the current results) in those with high HADS scores (i.e., rostral LC indicators may reflect stress/anxiety).

We thank the reviewer for pointing out these interesting findings. We included the reference in the revised version of the manuscript.

Comment 3. How was "more clustered functional organization" computed? I could not find description of this in the analysis section. If it is something that is evident from the visual depiction of the surface rendering shown in Fig. 2, please explain as it was not clear to me.

We thank the reviewer for the comment. As mentioned in a previous answer to a comment made by the Reviewer, clusterability was quantified in the following way (based on Ngo et al. 2021 NeuroImage). Gradient maps were clustered into k=2 clusters using the k-means clustering algorithm, then the Calinski-Harabasz criterion was calculated for each individual gradient map, which was then used as a measure of clusterability. Higher criterion values were significantly associated with older age (Spearman’s rho = 0.3129, p<0.0089), indicating that the gradient was more clustered in older individuals. This analysis is now included in the revised version of the manuscript.

Comment 4. In the connectopic mapping methods, it is stated that the analysis starts by calculating functional connectivity matrices between all voxel time series in an ROI and time series from a target mask. That statement sounds as though there would be one time series from the overall target mask. It is then stated that the target mask consistent of brain areas from a cortical and subcortical parcellation. But it is not clarified if (as I assume was the case) time series were extracted for each parcel within the mask (and how many parcels there were - 180?).

We thank the reviewer for the helpful comment. Regarding the target mask, average time series were extracted from each parcel in the atlases separately, and then pairwise correlations were calculated with timeseries from all voxels in the ROI (the LC). We used the Glasser-atlas (which contains 360 parcels) as a cortical parcelllation and the Tian-atlas (which contains 50 parcels) as a subcortical parcellation. The corresponding section of the manuscript now includes this clarification.

Comment 5. Then it is stated that "Afterwards, we obtained a similarity matrix from the functional connectivity matrices of LC ROI voxels by calculating the eta-squared measure." It would help here to explain a little more to clarify which things are being compared for similarity. Specifically, for which pairs was the eta-squared measure computed for?

The eta-squared measure was calculated between the functional connectivity profiles (or “fingerprints”) for all pairs of voxels in the LC ROI. More specifically, one such fingerprint contains the Pearson correlation coefficients between a given LC voxel time series and the regional time series from the target mask. The similarity matrix contains the eta-squared similarity of these fingerprints, therefore one index in the similarity matrix contains the similarity between the fingerprints of two specific LC voxels. The corresponding section of the manuscript now includes this clarification.

Comment 6. In Fig. 3, I found the labeling of surface renderings confusing (i.e., did high->low apply to both rows? What about 'emotional memory? do the top and bottom row correspond with the R/L LC?).

We thank the reviewer for the helpful comment and made some changes to Fig. 3 to clarify the labels. The upper row shows the right LC, whereas the bottom row shows the left LC. High->low and low->high applies to both rows. Regarding emotional memory, a worse performance on this task resulted in lower scores. With emotional reactivity, higher scores indicate a worse ability to regulate negative ratings on the task, which results in an inverse relationship of this score with the LC gradient features. We also extended the figure label to include this explanation.

**Response to Reviewer #2 (Public Review):**
Comment 1. One of the major strengths in the current study is the implementation of the fully data-driven, gradient-based method for mapping connectopies of the LC. This approach is especially suited for brain structures that are difficult to localise because the resulted connectopic mapping is relatively robust to ROI definition (Fig. 7 in Haak et al., 2018). However, as a very inclusive definition of the LC (the "meta atlas") was adopted in the study, to what extent the gradient approach can tolerate changes of accuracy and specificity for LC ROI definition is unknown. Some comparative analyses would be helpful to provide assessments on the specificity and stability of the reported gradient pattern.

We thank the reviewer for the positive assessment of our manuscript. Indeed, an advantage of the connectopic mapping approach is that it is less sensitive to minor ROI inaccuracies, which is convenient for the LC. We repeated the gradient calculation using a larger LC mask from Tona et al. (2017), and included a supplementary figure (Figure S3) that shows how the gradients still retain their rostrocaudal pattern using both LC masks.

Comment 2. Haak et al. showed distinct reproducibility within and between subjects when comparing connectopic mappings between M1 and V1. M1 connectopic mapping showed very high consistency across subjects (ICCs > 0.9) compared with V1. This is very reasonable because the functional organisation within M1 is relatively homogeneous. Regarding the reliability of the LC rostro-caudal gradient, the authors only stated that "individual gradient estimation is often not consistent", but direct measurement on the consistency across subjects for the LC gradient was missing. This is important for future LC fMRI studies as more consistent pattern might warrant the application of an atlas-based method otherwise a more individualized pipeline is needed for investigating functional dissociation in LC subregions.

We thank the reviewer for the question. Indeed, investigating the replicability of gradients at the individual level is important. However, regarding the LC, because of the ROI size and the relative shortness of the scans in the Cam-CAN dataset, we did not calculate individual level gradients and resorted to a group-level approach as we described in the method. Therefore, the assessment of individual reliability was outside of the scope of the current study. We included this as a limitation in the Discussion of the revised manuscript.

Comment 3. It puzzles me that why a dichotomous rostral vs caudal comparison was used to demonstrate the difference in connectivity patterns along the rostro-caudal gradient which might be an oversimplistic approach as described by the authors themselves? In fact, it might be more interesting to include the central "core" LC which is structurally organized in high density (Fernandes et al., 2012) and functionally distinguishable to the peri-LC "shell" region (Totah et al., 2018; Poe et al., 2022).

We thank the reviewer for the comment. Indeed, during the analyses we tried to delineate a central core region within the LC, however, the functional connections in this region varied greatly between individuals and we failed to reliably detect a functionally distinct central core region using FC. One reason for this might unfortunately be the limited spatial resolution of functional MRI. Instead, we hypothesized that the gradient manifests in fMRI connectivity of the LC by a gradual transition of connectivity profiles between the two dominant extremes of the caudal and rostral LC and we aimed to depict these two extremes in Figure 1. Although it is a simpler approach compared to the results of histological studies, we demonstrate in the paper that it still provides valuable information about LC in aging and LC-related behavioral measures.

Comment 4. The composition of rostral vs caudal connectivity pattern changes over ageing, where the loss of rostral-like connectivity was consistent in bilateral LC whereas the gain of caudal-like connectivity in older subjects was only evident in the left LC. Do authors have any explanations on this left-lateralised ageing effect which is interestingly coincided with a lot of observations such as increased left LC contrast ratios was found during ageing (Betts et al., 2017) and in PD patients (Ye et al., 2022), reduced left LC-parahippocampal gyrus connectivity was reported in aMCI patients (Jacobs et al., 2015).

We thank the reviewer for the question. Indeed, we observed lateralized changes in the LC gradients both in connection with aging and cognitive performance. Generally, the LC connects to several highly lateralized cortical networks, e.g. the salience and frontoparietal networks, which might result in an asymmetric plasticity in the LC. Interestingly, neurodegenerative disorders seem to affect the left LC more, e.g. more widespread loss of connectivity between the left LC and resting state networks was found in PD patients, with a correlation between left LC-executive control network connectivity and cognition (Sun et al. 2023). However, the biological basis for this is elusive, as post-mortem studies generally find the bilateral LC symmetric and mostly report pathological changes in the rostral and middle LC (Beardmore et al. 2021). In our case, a possible interpretation is that with the loss of rostral-like connectivity or previously rostral-like areas lose their specific connections and become more similar to the caudal part in terms of connectivity. In our study, since we did not investigate the cerebellum and the spinal cord, the typical caudal connectivity profile is more non-specific, since some of its dominant connections are not assessed. This interpretation is now included in the revised version of the manuscript.

**Reviewer #2 (Recommendations For The Authors):**
Comment 1. Minor:the preprocessing pipeline for HCP 7T data was not reported.

We extended the details of the preprocessing pipeline for the HCP 7T dataset.

Comment 2. - a difference map would be useful to demonstrate the similarity of LC connectivity gradient between CamCAN and HCP dataset.

We have now added a difference map between the CamCAN and HCP gradient in the supplementary material (Figure S2).

Comment 3. - labels for left and right LC were missing in Fig 3.

We corrected the labeling in Figure 3.

Comment 4. - in Statistical Analysis, CamCAN participants were divided into two groups with and without depressive and anxiety symptoms. It is unclear whether participants with high HADS scores were presented with both symptoms or just one of them.

Because of the low number of participants with high depression scores on the HADS test, we defined high HADS scores as individuals scoring above normal on either the anxiety part, the depression part, or both.